# Clinical Evaluation of Resin Composite CAD/CAM Restorations Placed by Undergraduate Students

**DOI:** 10.3390/jcm10153269

**Published:** 2021-07-24

**Authors:** Valentin Vervack, Peter De Coster, Stefan Vandeweghe

**Affiliations:** Reconstructive Dentistry, Dental School, Ghent University Hospital, Entrance 25, 9000 Ghent, Belgium; Peter.DeCoster@ugent.be (P.D.C.); stefan.vandeweghe@ugent.be (S.V.)

**Keywords:** computer-aided design, dental restorations, permanent, undergraduate, students

## Abstract

To evaluate the clinical outcomes of resin composite CAD/CAM restorations in a prospective cohort study, and to assess patient and operator satisfaction after restoration placement, 59 indirect resin composite were placed by supervised undergraduate students, of which 43 restorations were followed over a mean period of 28 months (14–44 months) and evaluated using USPHS criteria. Patient and operator satisfaction levels were assessed using a visual analogue scale (VAS) after restoration placement. A total of 37 patients and 47 restorations were included for further study. Four teeth were extracted—three due to extensive drug-induced secondary caries in the same patient, and one tooth due to large periapical periodontitis after 44 months of service. The overall survival rate was 91.4%, and success rate was 87.2%. Differences between baseline and endpoint scores were significant for marginal discoloration (*p* < 0.05) and adaptation (*p* < 0.001). Color match (*p* < 0.05) and surface texture (*p* < 0.001) differed significantly, affecting all restoration types. VAS scores for patient and operator satisfaction showed a significant rank correlation (*p* < 0.01), and pairwise comparison showed significant differences for mean overall patient and operator VAS scores (*p* < 0.001). Lava Ultimate CAD/CAM may be considered a suitable material for overlays and endocrown restorations when combined with IDS, air abrasion, and MDP-containing adhesive systems. Marginal disintegration may present in inlays and onlays over time.

## 1. Introduction

For many clinicians, direct composite resin restorations are the first choice when treating decayed teeth. A number of technical limitations, such as anatomical challenges in teeth with major substrate loss and marginal leakage associated with deep proximal boxes, as well as the often disappointing lifespan of composites, have legitimized the indirect restorative approach combining extra-oral fabrication and the use of materials with superior mechanical properties. Over the past decades, monolithic computer assisted design–computer assisted machining (CAD/CAM) restorations have gained popularity and have started to replace large direct composite buildups. The digitalization, and more specifically the introduction of chairside CAD/CAM fabrication units, has made the workflow for the manufacturing of indirect restorations easier, faster, and more accurate compared to the conventional workflow using impressions and cast models [1]. Evidence has been produced that a full-digital workflow offers more accurately fitting intracoronal restorations and complete crowns as well as fixed partial dentures [2,3,4]. According to Nedelcu et al. [5], digital data capturing can reliably replace conventional impression-taking when restoring up to ten units.

In terms of efficiency, the digital workflow offers some distinct advantages over the conventional workflow. Intra-oral scanning (IOS) and digitization of the dentition and the subsequent creation of a virtual cast eliminates two steps of the conventional workflow: the conventional impression and the fabrication of the gypsum cast. This simplification adds to the accuracy of the (virtual) master model but also reduces the working time. Overall, a digital workflow requires less chairside time for the dentist and less working time for the dental technician as compared to the conventional workflow [6,7,8]. Nevertheless, the learning curve of the IOS procedure represents a possible limitation for competent use. Recent studies verified that learning curves are influenced not only by individual factors such as previous experience and motivation, but also by the IOS system itself and repetition of practice [9,10]. Marti et al. [11] reported that the learning rate with older IOS devices was longer and led to a less positive student attitude towards digital scanning than with recent devices. According to Al Hamad et al. [9], a learning phase of five trials was needed to achieve a competence of 80% of the practitioner’s best performance. The scanning time and difficulty level decreased with the repetitive use of IOS. Other authors found that students roughly take the same amount of time for digital impressions but take significantly more time for a conventional impression in comparison with graduated dentists [12,13,14,15]. Furthermore, the learning curve for dental students can be different depending on the used IOS system [16,17].

Routinely used chairside CAD/CAM materials for monolithic restorations include glass ceramics and metal-oxide ceramics. Lithium disilicate reinforced glass ceramics present a three-point flexural strength around 350 MPa and offer natural esthetics due to favorable translucency values [18]. These materials have demonstrated excellent clinical survival rates between 83.5% and 99% after 10 years, depending on the type of restoration and the location in the dental arch [19,20,21]. Metal-oxide ceramics, more specifically yttria-stabilized tetragonal zirconia, offer an even higher flexural strength depending on their composition, ranging up to 1450 MPa, and can therefore be reliably used for posterior complete crowns as well as for long-span fixed partial dentures [22,23]. More recently, most manufacturers have expanded their range of zirconia materials by increasing the yttria content up to 5 wt%, thereby increasing the cubic crystal phase to 50% and higher [24]. Although this microstructural modification has reduced the material’s fracture toughness, it has improved the optical properties, namely in obtaining a degree of translucency close to glass ceramics [25,26].

In recent years, new resin-based CAD/CAM materials have been introduced, which combine the flexibility and ease of use of resin with the durability and surface stability of ceramic materials [27]. It was reported that these resin-containing CAD/CAM materials, referred to as hybrid ceramics (e.g., Enamic) or resin composite (e.g., LAVA Ultimate and Cerasmart), cause less wear on the opposing dentition compared to glass-ceramics and are easier to mill, polish, and adjust [28,29]. In addition, the marginal integrity is superior to that of glass ceramics due to a lower brittleness [30]. Resin composite CAD/CAM restorations present a better overall mechanical performance than conventional resin composite due to their higher degree of polymerization [31,32]. Unlike indirect resin composites, direct resin composites often suffer from marked polymerization shrinkage and stress, which may cause enamel cracks, debonding of the hybrid layer and thus post-op hypersensitivity and a higher risk for secondary caries [33,34]. These new materials also offer a more efficient fabrication than their ceramic homologues since less milling time, fewer milling tools, and no post milling furnace firing are needed.

Lava Ultimate (3M Espe, Seefeld, Germany) was one of the earliest introduced monolithic resin composite CAD/CAM materials with a total nanoceramic material content by weight of approximately 80% and a flexural strength of 200 MPa. Its zirconia-silica nanocluster particles are synthesized via a proprietary process from 20 nm silica particles and 4–11 nm zirconia particles, producing an average nanocluster particle size of 0.6–10 micrometers. The indications include veneers and intra-coronal restorations, such as inlays, onlays, and overlays. The number of clinical studies involving Lava Ultimate restorations, however, is presently limited. In a recent split-mouth study, Souza et al. [35] reported a 100% clinical success rate of both lithium disilicate and Lava Ultimate onlays over a 1-year period. When comparing Lava Ultimate and direct composite resin restorations outcomes after 2 years of service, Tunac et al. [36] confirmed a 100% retention rate and no significant differences between the materials in any of the clinical criteria. Fasbinder et al. [37] reported a Kaplan–Meier probability for onlay fracture of 0.083 (CI 0.036; 0.189) after 5 years, which was not significantly different from leucite-reinforced ceramic.

Although CAD/CAM resin composites have been recommended for indirect restorative treatment without entailing failures intrinsic to all-ceramic materials, such as chipping, concerns have been raised about their clinical performance. Debonding and cohesive fractures have been reported by previous authors and have been linked to the material’s resilience, leading to a revision of the clinical indications in 2015. Although Lava Ultimate has been used for over 7 years, independent studies evaluating the clinical performance are scarce [35,36,37,38,39]. Therefore, the purpose of this clinical trial is to evaluate the mid-term outcomes of resin composite intracoronal chairside CAD-CAM restorations up to 44 months using internationally accepted USPHS standards. The study also aims at evaluating the patient and operator satisfaction immediately after placement of the restoration.

## 2. Materials and Methods

A prospective clinical study was designed in compliance with the Declaration of Helsinki, and the study protocol was approved by the Ethical Committee of the Ghent University Hospital (2015/0144). Patients aged 18 years and older requiring restoration of a decayed molar or premolar involving 3 surfaces or more were recruited in the dental clinic of the Ghent University Hospital. Patients with parafunctions and high caries risk and teeth that are in need of full crown preparations were excluded from the study. All patients were treated by undergraduate dental students (4th and 5th year of dental school) under supervision of an experienced dentist. All students already received preclinical training in indirect restorative dentistry as part of their dental education and an additional practical training in digital impression taking. Patients were informed about the study protocol and had to provide written informed consent prior to enrollment in the study.

After caries removal or elimination of defective restorative material, minimal tooth preparation was performed to maintain as much sound tooth structure as possible. Involved weakened cusps (less than 1.5–2mm thickness) were reduced by 1.5 mm; the butt joint margins were designed to be sharp, and the internal line angles were rounded. Immediate dentin sealing (IDS) was applied under a rubber dam using a 2 step self-etch adhesive system (Clearfill SE bond, Kuraray, Tokyo, Japan) and a flowable composite liner (SDR, Dentsply-Sirona, St. York, PA, USA). Deep subgingival margins were elevated, and undercuts were filled using a micro-hybrid direct composite (Clearfill AP-X, Kuraray, Tokyo, Japan). The remaining oxygen inhibited layer was covered with glycerin gel and light cured. The enamel was cleared of any adhesive using a fine diamond bur. Contrast powder was applied, and a digital impression was made using an intra-oral scanner, based on the principle of active wavefront sampling (True Definition, 3M Espe, Seefeld, Germany). A provisional restoration was made chairside from a bis-acrylic composite (Protemp 4, 3M Espe, Seefeld, Germany), based on a putty taken before start of the treatment (Exaflex^®^ Putty, GC, Tokyo, Japan) and fixated using a temporary cement (Rely X Temp, 3M Espe, Seefeld, Germany). The digital impression was sent to a milling center (DPI Lava Milling Center, Anderlecht, Belgium), where the restoration was designed and milled from a nanoceramic particle reinforced resin composite block (Lava Ultimate, 3M Espe, Seefeld, Germany). The unfinished restoration was sent back to the dental school, where it was polished and stained (Sinfony, 3M Espe, Seefeld, Germany) by the undergraduate students.

Prior to the final cementation, the temporary restoration was removed, and the preparation was sandblasted using 27 µm Al_2_O_3_ particles (Rondoflex, Kavo, Biberach an der Riss, Germany) to remove cement remnants and to reactivate the IDS layer. The enamel was etched using phosphoric acid 37% for 20 s, and a universal adhesive (Scotchbond Universal, 3M Espe, Seefeld, Germany) was applied and left uncured. The intaglio of the restoration was sandblasted using 27 µm Al_2_O_3_ particles and a universal adhesive (Scotchbond Universal) was applied without light-curing. The restorations were then luted using a dual-cure adhesive resin cement (Rely X Ultimate, 3M Espe, Seefeld, Germany). After seating, tack-curing was performed for 2 s, whereafter the cement overflow was removed using a sharp scaler. After removal of the cement remnants, the margins were polished using a fine diamond bur and polishing rubbers. All clinical steps were performed using rubber-dam isolation.

One independent evaluator was calibrated and tasked to examine all restorations in the study. Clinical assessments were made at baseline (one week after placement) and at follow-up examination sessions using modified USPHS criteria (Table 1) for retention, color match, marginal discoloration, marginal adaptation, secondary caries, anatomical form, and surface texture. Periapical radiographs were taken to verify the correct seating of the restoration and to detect secondary caries at follow-up examinations. Vestibular, lingual, and occlusal views were documented using clinical pictures.

The restorative outcome was defined in terms of restoration success, restoration survival, and tooth survival. Restoration success means that no reversible or irreversible complications occurred to the restoration or the tooth. Restoration survival means that reversible complications occurred over time, but it also means that these could be repaired. Complications included chipping, minor fractures, and debonding. Tooth survival means that the tooth was still present at the time of evaluation. In the case of loss of the restoration, the tooth could be restored with a new direct or indirect restoration.

Patient and operator satisfaction were assessed using a visual analogue scale (VAS) questionnaire (see Table 2, recorded as a VAS score on a line from 0 to 100 mm, with 0 mm being very bad, 100 mm being excellent). Patients were asked to mark for each question the respective VAS, which was a 100 mm straight horizontal line with the left end indicating “not at all satisfied” and the right end “very satisfied”. The satisfaction value was determined by the distance from the left end of the scale to the mark in millimeters and expressed as percentage (10 mm corresponds to 10%, 20 mm 20%, etc.). The aspects addressed in the patient satisfaction questionnaire were the IOS procedure, the esthetic outcome, and the functional comfort provided by the restoration. Operator satisfaction was rated with respect to the overall restorative procedure, the IOS procedure, the final design of the restoration, the ease of placement of the restoration, and the result in terms of esthetics, color, and shape of the restoration. In addition, the students were asked to rate their satisfaction with the digital workflow as compared to the conventional workflow.

A Wilcoxon signed-rank test for dependent samples was used to compare the baseline and follow-up USPHS criteria and also to compare the mean VAS scores between patients and operators. A Kruskal–Wallis test with post-hoc pairwise comparison and Bonferroni correction as well as Pearson’s Chi-squared test were used to analyze the statistical differences between the USPHS criteria of the restorations, grouped as onlays (incl. inlays), overlays, and endocrowns. Spearman’s coefficient indicated the rank correlation between patient and operator satisfaction. All statistics were performed using IBM SPSS Statistics 27 (IBM Corporation, Armonk, NY, USA)

## 3. Results

A total of 45 patients were enrolled in the study: 17 males and 28 females with a mean age of 48 ± 13 years (range 21–83 years); 59 restorations were initially placed in 43 molars and 16 premolars, including 27 overlays, 16 endocrowns, 12 onlays, and 4 inlays. Out of all patients, 11 of them with 12 restorations did not return for their follow-up examinations and were considered as drop-outs. Four teeth were extracted—three due to extensive drug-induced secondary caries in the same patient, and one due to large periapical periodontitis. In total, 34 patients and 43 restorations were included in the study for qualitative assessment. Figure 1 illustrates the flow of patients and restorations enrollment for both the evaluation of restoration survival and the qualitative assessment by using USPHS criteria. Table 3 and Table 4 display the distribution of the location of the included restorations and the restoration types.

An overall restoration survival rate of 91.4% was found after a mean follow-up of 28 ± 8 months (range 14–44 months). Three onlays were lost for follow-up due to extraction following rampant drug-induced cervical caries in one patient. One tooth with a fractured endocrown and a large periapical lesion was extracted. The overall restoration success rate was 87.2%. In one patient, two teeth with an overlay developed secondary caries, which was treated by removing the decay and placing a direct composite filling without removing the onlays. Figure 2 presents the number of USPHS-scored restorations per time frame.

In Figure 3, an example case of 36 months follow-up on the lower left first molar onlay is shown. Although no marginal discoloration is visible, a marginal gap has formed on the occlusal margin after 36 months and was rated as Bravo. The follow-up time length on the lower left second premolar is 27 months. Here, a small occlusal marginal gap was reported and scored as Bravo for marginal adaptation; both onlays received a Bravo score for surface texture due to loss of luster. The baseline and endpoint scores of the USPHS categorical criteria are shown in Table 5. The scores remained relatively unchanged for all restoration types at a ≥95% Alpha over an averaged 28-month period for retention, secondary caries, and anatomic form. In terms of color match, 49% of restorations were scored as Alpha, 42% as Bravo, and 9% as Charlie at baseline. Differences between color match USPHS scores at baseline, and at endpoint they were statistically significant (Wilcoxon’s test; Z = −2.840; *p* < 0.05). Marginal discoloration scores dropped from 100% Alpha at baseline to 84% Alpha and 16% Bravo at endpoint, which was predominantly registered in onlays (Z = −2.646; *p* < 0.05). With respect to marginal adaptation, the baseline 100% Alpha score dropped to 72% and a 28% Bravo score, which was again predominantly related to onlays and, in a lesser degree, to overlays. Scores at baseline and at endpoint were significantly different (Z = −3.464; *p* < 0.001). Surface texture showed a statistically significant difference between the baseline (100% Alpha) and endpoint (53% Alpha and 47% Bravo; Z = −4.472; *p* < 0.001), affecting all three restoration types. In general, endocrowns presented the smallest drop of categorical scores over the observation period, and inlays and onlays the greatest. An independent samples Kruskal–Wallis test showed statistically significant differences between restoration types for marginal discoloration (H = 13.675; *p* < 0.05) and marginal adaptation (H = 25.124; *p* < 0.001). When comparing restoration types pairwise, inlays and onlays showed a significantly greater drop for marginal discoloration than overlays (H = 9.726; *p* < 0.05) and endocrowns (H = 10.750; *p* < 0.05) Marginal adaptation scores were significantly lower for inlays and onlays compared to overlays (H = 15.869; *p* < 0.001) and endocrowns (H = 17.917; *p* < 0.001). Chi-square test results are displayed in Table 5, showing the same outcomes as the above described tests.

The median and interquartile range of patient and operator satisfaction scores are depicted in Figure 4. The overall mean VAS score for patient satisfaction was 86.5% ± 10.2% versus 77.0% ± 10.6% for operator satisfaction. The mean VAS score for patient satisfaction with the IOS procedure was 77.1% ± 24.7%, while satisfaction with the esthetic result and functionality of the restoration were 92.0% ± 7.0% and 90.6 % ± 8.9%, respectively. For operator satisfaction, a mean VAS score of 70.7% ± 18.7% was calculated with the overall digital workflow, 71.7% ± 24.1% with the IOS procedure, 75.9% ± 21.5% with the virtual designing of the restoration, 80.8% ± 15.6% with the placement of the restoration, 83.7% ± 12.7% with the esthetic appearance, 76.4% ± 17.3% with the color, and 83.0% ± 16.9% with the anatomical shape of the restoration. In addition, 95.6% of the operators preferred the IOS procedure over conventional impression taking.

A statistically significant correlation was found between mean patient and operator satisfaction (Spearman’s ρ = 0.609; *p* < 0.01). The mean overall VAS scores showed a statistically significant difference between patients and operators (Z = −4.375; *p* < 0.001).

## 4. Discussion

The present prospective study evaluated the survival, success, and clinical performance of Lava Ultimate restorations, and the patient and operator satisfaction related to the restorative intervention. All restorations were prepared and installed in an academic setting by undergraduate students who received previous training. To the authors’ knowledge, no study to date has analyzed the clinical outcomes of Lava Ultimate over a period longer than two years, and no satisfaction reports have yet been published on this treatment approach. In the clinical report of Zimmerman et al. [39], a clinical success rate of 95.0% after 12 months and of 85.7% after 24 months was determined, with debonding as most prominent complication. Our results confirmed a clinical success rate of 87.2% after a mean 28 ± 8 months follow-up period, with secondary caries as the only complication. Although several authors related debonding to the physicochemical properties of Lava Ultimate, no such complications were registered in our patient sample. In this respect, previous studies have reported on the effect of material conditioning for optimizing the adhesive behavior of particle-filled composite resin. In an in vitro setting, Frankenberger et al. [40] evaluated the microtensile bond strength of Lava Ultimate, e.Max CAD, Celtra Duo, and Vita Enamic, and they concluded that sandblasting without application of hydrofluoric acid and silane produced the highest bond strength values for Lava Ultimate (17.9 ± 4.5 MPa), but this is still inferior to those of lithium disilicate ceramic (26.3 ± 7.7 MPa). One possible explanation for this is, since no ceramic scaffold is available for microretentive anchorage, particle-filled composite resin materials might be more susceptible to debonding. Rosentritt et al. [41] reported a high incidence of debonding of Lava Ultimate restorations and postulated that this might be attributed to a swelling of the restoration after water absorption in combination with deformation of the highly elastic material under occlusal loading. Lava Ultimate thus seems to have weaker mechanical properties than other materials such as glass ceramic alternatives, entailing more complications over extended periods in function.

Several factors might influence restoration survival and success, including patient-related caries risk and occlusal load [42,43,44]; in addition, the clinical experience of the operator may contribute to differences in treatment outcomes. The latter involves not only the designing of the preparation, the execution and accuracy of the IOS, but also the cementation procedure and finishing of the restoration [45]. In order to limit the influence of these factors, patients with high caries risk were excluded from the study at baseline. The abutment preparation and the designing and finishing of the restoration were carried out according to the prevailing guidelines as taught in the undergraduate training. On the other hand, patients with parafunctional habits or heavy wear facets were not excluded from the study, which probably may have affected the clinical outcomes of the restorations.

In contrast to most clinical studies, the restorations were placed by undergraduate students exclusively, albeit after thorough training and supervised by calibrated clinical instructors. Prior to the start of the study, all operators received clinical IOS training. As confirmed by previous authors, repeated clinical training before the actual treatment decreases the time needed for a digital full arch scan and significantly improves the accuracy of the digital impression [46,47]. In a similar study setup, Zimmerman et al. [48] reported that almost all undergraduate students (95%) wanted the CAD/CAM method to be integrated in their regular courses, which is in line of our poll among our operators (95.7% chose digital as preferred impression method).

In some cases, minor adjustments at the occlusal and approximal surface were done to ensure a correct seating and occlusal contact. Discrepancies in the fit and occlusal and approximal contacts could be caused by errors in the digital impression, attributable to the type and calibration of the device, the IOS method, and operator experience [12]. The accuracy of IOS also depends on the extent of the scanning region and decreases when the scanning surface increases [49]. In spite of these adjustments, the placement of the restoration was rated a mean VAS score of 80.8% ± 15.6%, which was in line with the report of Joda et al. [50], confirming that the digital workflow entailed less adjustments than the conventional workflow. The high accuracy and reproducibility of the IOS method, the simplified CAD/CAM fabrication process, and the limited need for manual interventions, are likely to result in a higher output accuracy, thus allowing for less room for flaws or irregularities [50,51]. For future clinical studies, it could be recommended to run customized training programs for undergraduate students depending on the individual level of skill and the IOS system in use. This might reduce the level of difficulty and the scanning time and might increase the accuracy of the scans [9,10,11,12,13,15,16,17,46,47,48,50].

The clinical performance of Lava Ultimate restorations has previously been studied using either the USPHS [37] or FDI criteria [35,36,39,52]. The USPHS criteria used in this study do not fully comply with the modified set proposed by Fasbinder et al. [37], who added extra subset scores for most of the criteria except for color match and anatomic form. Our findings indicated a significant drop in USPHS scores at the endpoint for color match, marginal discoloration, marginal adaptation, and surface texture. Although the sample sizes of the restoration type subgroups were small, there was strong statistical evidence confirming the significant differences between inlays/onlays and the other restoration types. The color of the restorations mimicked the adjacent tooth structure (USPHS Alpha score) in 26% at endpoint, while 63% were found to have an acceptable mismatch (USPHS Bravo score), and 11% showed an unacceptable color mismatch (USPHS Charlie score). In the study of Zimmerman et al. [39], 13% of restorations mimicked the adjacent tooth color, and 87% showed acceptable minor or clear discrepancies in color match. Souza et al. [35] assigned a good color match to 25% of Lava Ultimate restorations and a minor or distinct but acceptable color deviation with the adjacent tooth structure to 75%, which was better than the IPS e-max CAD onlays in a split-mouth design. Only Fasbinder et al. [37] reported an almost ideal (93%) match between tooth and restoration after 5 years, which was comparable to leucite reinforced ceramic. Lava Ultimate blocks, however, are available in a limited range of monochromatic colors. The often-marked shade gradient of natural tooth surfaces and the absence of an esthetic bevel preparation for this type of restorative material may account for several color mismatches, although on the other hand, particle-filled composite materials appear to be more susceptible to discoloration compared to dental ceramics [28,30,53]. In addition, the FDI and USPHS criteria used for color evaluation lack objectivity and evaluator calibration, which could account for some distinct differences between studies.

Marginal discoloration without axial penetration (USPHS bravo score) was found in 16% of restorations, which was statistically significantly correlated with inlays and onlays. Zimmerman et al. [39] reported marginal staining in 60% of Lava Ultimate restorations after 24 months. Fasbinder found an unchanged marginal discoloration score over 5 years of over 93%, while Souza et al. and Tunac et al. reported an 85% and 96% fraction, respectively, showing stainless margins. To the best of our knowledge, Zimmerman et al. [39] did not use IDS and applied a dual cure adhesive from another brand rather than the restorative material (Variolink II, Ivoclar Vivadent AG, Schaan, Lichtenstein). Both Tunac et al. [36] and Souza et al. [35] used the 10-MDP containing Rely X Ultimate adhesive system (3M, St. Paul, MN, USA). Fasbinder et al. [37] compared the abovementioned luting cements but did not find any significant difference over a 5-year follow-up period. Based on the above and present findings, IDS in combination with a self-etching and 10-MDP containing universal adhesive and a dual cure composite cement after airborne particle abrasion of both the receiving abutment and restoration intaglio appears to produce the most favorable bonding performance. This was also suggested in the in-vitro study of Kömürcüoğlu et al. [54].

With respect to marginal adaptation, a detectable but clinically acceptable margin (USPHS bravo score) was found in 28% of the restorations, producing a statistically significant difference (Z = −3.464; *p* < 0.001) between the baseline and endpoint scores with a strong correlation to inlays and onlays compared to overlays (H = 15.869; *p* < 0.001) and endocrowns (H = 17.917; *p* < 0.001). Fasbinder et al. [37] reported detectable margins along less than 50% of cavosurface margin and less than 1 mm in depth after 5 years in 77% of Lava Ultimate restorations as opposed to 87% in glass ceramic restorations. This was most obvious at the occlusal margins, where the cement gap seems to wear faster than the restoration. Tunac et al. [36] reported small marginal gaps that are removable by polishing (150-micron grid) in 5% of Lava Ultimate inlays after 2 years of function. A harmonious outline without gaps was reported in 83% of restorations by Zimmerman et al. [39] and in 90% by Souza et al. [39]. Marginal integrity is one of the most important factors in rating a restoration’s success. Even in restorations with clinically acceptable margins, disruption of the marginal seal caused by material instability or disintegration of the luting cement, as previously documented in ceramic restorations, may occur over time. The low wear resistance and low stiffness of Lava Ultimate may expose restorations to marginal step formation, in contrast to the gap observed in ceramic restorations [35,42,55]. It follows that the marginal adaptation of Lava Ultimate restoration should be closely monitored during successive follow-up sessions.

In 44% of restorations, the surface texture was scored rougher than enamel, albeit clinically acceptable (USPHS Bravo score). Zimmerman et al. [39] reported a slight but polishable dullness in 50% of mixed-type restorations after 24 months. Souza et al. [35]. Found that only 10% of the restorations kept a high surface luster. Tunac et al. [36] found similar surface changes in only 2% of Lava Ultimate inlays, and Fasbinder et al. [37] reported a loss of surface gloss without affecting texture in 7% of Lava Ultimate onlays, which was equal to glass ceramic onlays as part of their study. Although high incidences of surface luster loss were reported in all available studies, none were reported to be clinically unacceptable. Koizumi et al. [56] postulated that surface roughness and related luster of resin composite indirect restorations might further be influenced by external factors, such as toothbrush abrasion.

To the best of the authors’ knowledge, this is the first study reporting patient and operator satisfaction with CAD/CAM-fabricated resin composite restorations and related clinical workflow using the visual analogue scale (VAS) questionnaire [57]. In this study, the VAS was selected since it appears to be significantly more sensitive to registering small nuances in comparison to scales with defined categorical response options (very satisfied, satisfied, not satisfied, etc.), as used by other authors [39]. Although the mean overall VAS scores differed significantly between patients and operators (*p* < 0.001), satisfaction on the IOS procedure was comparable (77.1% ± 24.7% for patients and 71.7% ± 24.1% for operators), whereas on the other hand, the esthetic result was scored significantly poorer by operators (i.e., 83.7% ± 12.7%, versus 92.0% ± 7.0% by patients). The latter was in line of the findings of Zimmerman et al. [48], and it most probably reflects the more critical attitude of dental professionals, who are additionally trained to detect small color differences under dental operatory light. The significant correlation between patient and operator VAS scores (*p* < 0.01) suggests that a number of factors such as, for instance, the location, stage of deterioration and color of the treated tooth, restoration type, and accessibility of the abutment margins in terms of sulcus widening and margin isolation, probably might have influenced the individual appraisal of both the treatment flow and the restorative outcome. Finally, 95.6% of the operators preferred digital over analogue impression taking, which was also consistent with the previous report of Zimmerman et al. [48].

Some limitations must be considered when analyzing the present findings. First, no control group was included in this prospective observational study where the CAD/CAM resin composite could be compared to another direct or indirect restorative material. The focus of interest was on the clinical behavior of the at-the-time novel material class of particle filled resin composite. Previous clinical studies involving a control group indeed failed to show significant differences between the tested materials [35,36,37]. Furthermore, valuable data were lost by a drop-out of 20.3% and loss by extraction of 6.7% of the original study population, producing rather small sample sizes but still with enough statistical power to test our hypotheses. In addition, the variation observed between the findings of different studies may be caused by the possible inclusion of patients with dental wear and parafunctional habits, the extension of the restorations, the number of operators and evaluators, and the luting cement and procedure applied.

## 5. Conclusions

Within these limitations, the Lava Ultimate CAD/CAM restorations exhibited good survival and success rates when placed by undergraduate students and combining IDS, sandblasting, and MDP-containing adhesive systems. Marginal disintegration may, however, present in inlays and onlays over time. Patient and operator satisfaction with IOS procedures and restorations was high, even though disagreement in satisfaction scores was found for esthetical appraisal of the finished restorations. Integrating the full digital workflow for the fabrication of partial indirect restorations in the undergraduate training program may represent an important asset based on a manageable learning curve and the ease and efficiency of the procedure.

## Figures and Tables

**Figure 1 jcm-10-03269-f001:**
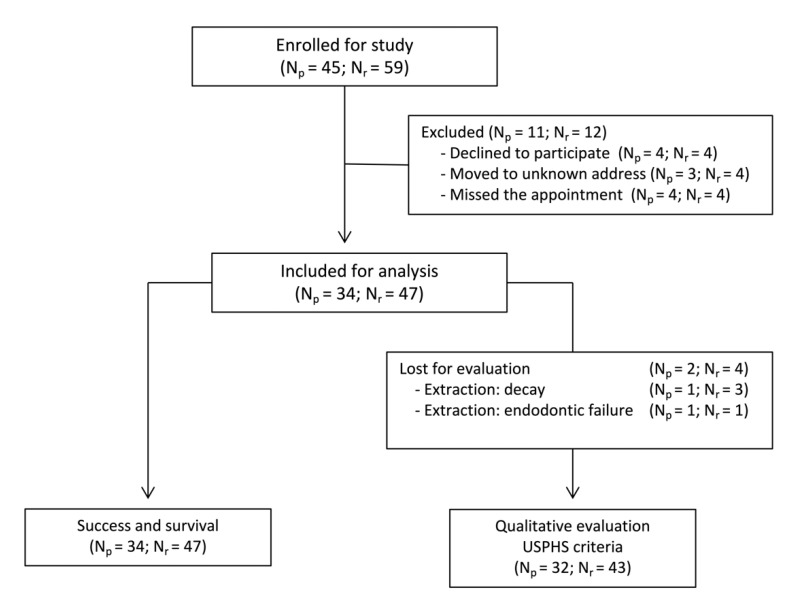
Flow chart (N_p_ = number of patients; N_r_ = number of restorations).

**Figure 2 jcm-10-03269-f002:**
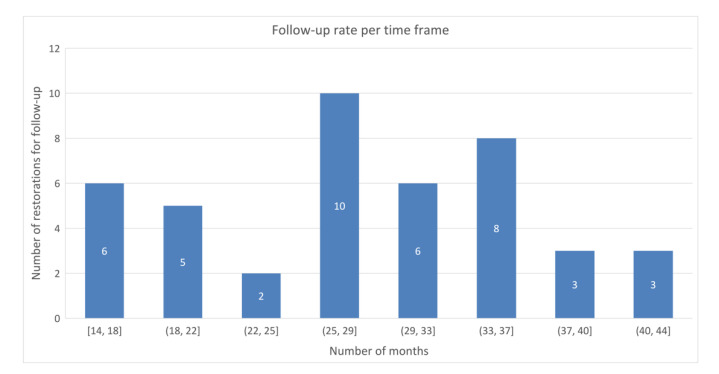
Follow-up rate per time frame.

**Figure 3 jcm-10-03269-f003:**
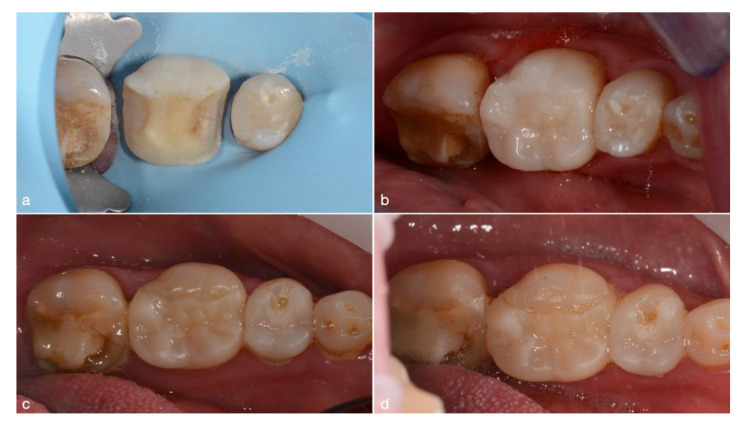
Images and descriptions of follow-up cases. Marginal adaptation: (**a**) Preparation at placement appointment; (**b**) immediately after placement of onlay on #36; (**c**) unintentional follow-up after 7 months of #36 due to placement of onlay on #35; (**d**) 36-month follow-up of #36.

**Figure 4 jcm-10-03269-f004:**
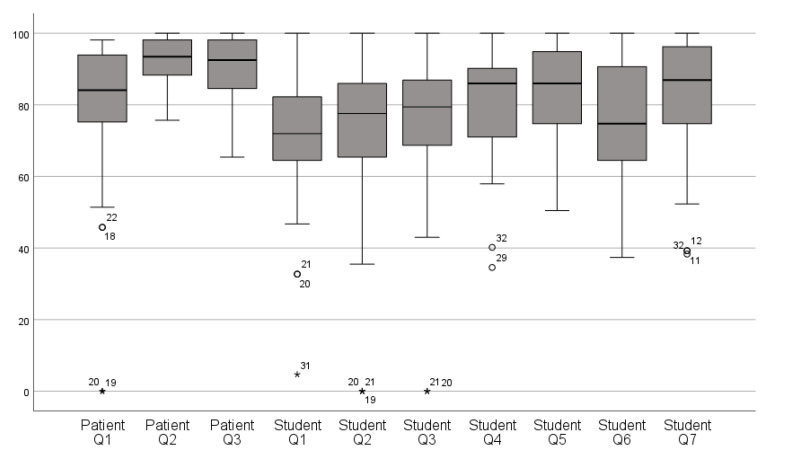
Boxplot with median and interquartile range of patient and operator VAS scores. (° = outliers; * = most extreme outlier; numbers = case numbers.)

**Table 1 jcm-10-03269-t001:** Modified USPHS criteria.

	Score	Criteria
Retention	Alpha	No loss of restorative material
	Charlie	Any loss of restorative material
Color Match	Alpha	Mimics tooth
	Bravo	Acceptable mismatch
	Charlie	Unacceptable mismatch
Marginal Discoloration	Alpha	No discoloration
	Bravo	Discoloration without axial penetration
	Charlie	Discoloration with axial penetration
Secondary Caries	Alpha	No caries present
	Charlie	Caries present
Anatomic Form	Alpha	Continuous
	Bravo	Slight discontinuity
	Charlie	Discontinous, failure
Marginal Adaptation	Alpha	Closely adapted, no detectable margin
	Bravo	Detectable margin clinically acceptable
	Charlie	Marginal crevice, clinical failure
Surface Texture	Alpha	Enamel like Surface
	Bravo	Surface rougher than enamel, clinically acceptable
	Charlie	Surface unacceptably rough

**Table 2 jcm-10-03269-t002:** Patient and practitioner questionnaire.

**Patient satisfaction**	1	How did you experience the intra-oral scanning procedure					
	2	How would you rate your final restoration in terms of					esthetics
	3						functionality
**Practitioner satisfaction**	1	How did you experience					the overall procedure
	2						the intra oral scanning
	3						the designing of the restoration
	4						the placement of the restoration
	5	How would you rate the overall esthetic appearance of the restoration					
	6	How would you rate the color of the restoration					
	7	How would you rate the shape of the restoration					
	8	Would you prefer digital or conventional impression taking?					

**Table 3 jcm-10-03269-t003:** Distribution of involved teeth (*N* = 43) in the study population.

Teeth	1st PM	2nd PM	1st M	2nd M	3rd M	Total
Maxillary	1	1	6	2	1	11
Mandibular	2	6	15	9	0	32
Totals	3	7	21	11	1	43

**Table 4 jcm-10-03269-t004:** Distribution of the involved restoration types.

	Premolars	Molars	Total
In-/onlay	4	8	12
Overlay	3	18	21
Endocrown	3	7	10
Total	10	33	43

**Table 5 jcm-10-03269-t005:** USPHS scores and Chi-Square statistics at baseline and endpoint per restoration type.

	Baseline					Recall Session				
Category	In/onlays	Overlays	Endocrowns	Total	*Chi-Squared*	In/onlays	Overlays	Endocrowns	Total	*Chi-Squared*
	*N*	*N*	*N*	*N (%)*	*p*	*N*	*N*	*N*	*N (%)*	*p*
Retention					*-*					*-*
Alpha	12	21	10	43 (100)		12	21	10	43 (100)	
Color match					*0.375*					*0.162*
Alpha	8	10	3	21 (49)		2	7	2	11 (26)	
Bravo	4	9	5	18 (42)		10	12	5	27 (63)	
Charlie	0	2	2	4 (9)		0	2	3	5 (11)	
Marginal discoloration					*-*					*<0.001*
Alpha	12	21	10	43 (100)		6	20	10	36 (84)	
Bravo	0	0	0	0		6	1	0	7 (16)	
Secondary caries					*-*					*0.333*
Alpha	12	21	10	43 (100)		12	19	10	41 (95)	
Charlie	0	0	0	0		0	2	0	2 (5)	
Anatomic form					*-*					*0.333*
Alpha	12	21	10	43 (100)		12	19	10	41 (95)	
Bravo	0	0	0	0		0	2	0	2 (5)	
Marginal adaptation					*-*					*<0.001*
Alpha	12	21	10	43 (100)		2	19	10	31 (72)	
Bravo	0	0	0	0		10	2	0	12 (28)	
Surface texture					*-*					*0.620*
Alpha	12	21	10	43 (100)		5	12	6	23 (53)	
Bravo	0	0	0	0		7	9	4	20 (47)	

## Data Availability

Data are available from the corresponding author upon request.

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
