# Peer review of "Clinical Evaluation of Resin Composite CAD/CAM Restorations Placed by Undergraduate Students"

_jcm, 2021, doi:10.3390/jcm10153269_

Round 1
Reviewer 1 Report
jcm-1273643-peer-review-v1
Up to 44 months clinical evaluation of nano-ceramic Computer Assisted Design/Computer Assisted Machining restorations placed by undergraduate Students
The aim of this study was to evaluate the clinical performance of Lava Ultimate restorations in a prospective cohort study, and to assess patient and operator satisfaction after restoration placement. Restorations were placed by different undergraduate students and were examined by one single evaluator applying USPHS criteria to 43 restorations in 34 patients.
- One of the most serious drawbacks of the study is that no control group was included. This circumstance was also not discussed. As a result, the study loses considerable significance.
- Please provide a reason why just 45 patients were recruited.
- The fact that 11 of the recruited 45 patients did not come for follow-up is a pity, but must be accepted as it is. The explanation why some restorations in the remaining 34 patients were not included in the analysis is not comprehensible.
- The patient is the unit of investigation in this study. It is therefore necessary to randomly select one restoration per patient and allow it for analysis.
- The flow of participants through the phases of the study is missing and needs to be presented urgently and with full details.
- There is now agreement in the literature that two evaluators independently assess the chosen criteria. In case of different evaluations, both then agree on a common value - still with the patient on the chair. Why was this not done?
- The authors mention at various points in the manuscript that the statements made are sufficiently reliable, but refrain from a power analysis. Please do an adequate power analysis and report these results.
- The first follow-up examinations took place after 14 months, the latest after 44 months. Considering that in similar studies follow-up intervals of 6 or 12 months are usual, this range is much too long. Appropriate periods should be defined here, e.g. first year, 2nd year, etc. and the data analysed accordingly.
- “Up to 44 months ….” in the title is completely misleading.
- Statistical evaluation
- A normality test on ordinal scaled USPHS data is false.
- Present all USPHS data as frequencies including appropriate percentages.
- Apply chi-square tests on USPHS data.
- The VAS results clearly show that these are not symmetrically distributed. Therefore, they cannot be normally distributed either. So it is necessary to present and evaluate these data non-parametrically. The representation as in Fig 2 is incomplete.
- Include manufacturers informations, Lot numbers, and all further details throughout the manauscript.
- The parts of the manuscript do not coincide, e.g., information to be presented in M&M is presented only in the Discussion (e.g.: exclusion of patients with high caries risk). Structure the manuscript very clearly.
- The unit of investigation was not chosen correctly. Therefore, it is not possible to review the results and their discussion.
- The discussion of the material used and the methods applied is not exhausting.
Author Response
Response to reviewer 1
To Reviewer # 1: First of all, we want to thank you for your insightful and useful comments. We tried to address all your comments in the best way we could, and we hope that the reviewed version of the manuscript has improved. We are still fully available to modify the text if again prompted.
- One of the most serious drawbacks of the study is that no control group was included. This circumstance was also not discussed. As a result, the study loses considerable significance.
Indeed, we agree that a split-mouth study design including a control group consisting of e.g. direct composite restorations (Tunac et al. 2019), leucite-reinforced (Fasbinder et al. 2019) or lithium disilicate glass ceramics (Souza et al.2021) would have provided relevant additional information on the clinical performance of the material in focus. The present study, however, was set in an educational environment involving a limited number of both undergraduate students and eligible patients. Part of focus of the study was on the ease and efficiency of CAD/CAM indirect restoration placement by undergrad student who have comparatively less clinical experience. On the other hand, we feel that the results do not necessarily loose significance in the absence of a control group (like Zimmerman et al. 2018) since comparison with outcomes published by other authors was made and addressed in the Discussion section.
To comply with your comments, we adapted the Discussion section as follows:
“Some limitations must be considered when analyzing the present findings. First, no control group was included in this prospective observational study, comparing CAD/CAM resin composite to another direct or indirect restorative material. The focus of interest was on the clinical behavior of the at the time novel material class of particle filled resin composite. Previous clinical studies involving a control group indeed failed to show significant differences between the tested materials [32-34]. Furthermore, valuable data have been lost by drop-out of 20,3% and loss by extraction of 6,7% of the original study population, producing rather small sample sizes, however with enough statistical power to test our hypotheses. In addition, the variation observed between the findings of different studies may be caused by the possible inclusion of patients with dental wear and parafunctional habits, the extension of the restorations, the number of operators and evaluators, and the luting cement and procedure applied.”
- Please provide a reason why just 45 patients were recruited.
Within the constraints of the educational setting, this was the number of undergraduate students who treated the patients. Each student was allowed to treat one patient.
- The fact that 11 of the recruited 45 patients did not come for follow-up is a pity, but must be accepted as it is. The explanation why some restorations in the remaining 34 patients were not included in the analysis is not comprehensible.
For a better understanding, we added a flow chart (figure 1) of the patients and restorations included, as well as the reasons for and numbers of patients/restorations that were lost for evaluation.
- The patient is the unit of investigation in this study. It is therefore necessary to randomly select one restoration per patient and allow it for analysis.
Selecting only one restoration per patient would presumably have simplified the statistics. However, for most material- and restoration-related variables, the restoration is considered as the unit of investigation as shown in previous articles that were referenced in the Discussion section (Zimmerman 2018; Tunac 2019;Fasbinder 2019; Souza 2021). Therefore we used this investigative unit for evaluation of USPHS scores. Since all restorations were made from the same material and placed following the same protocol, the clinical effect of the patient-related factors on the restoration’s margins etc., scored according USPHS criteria, could be esteemed limited. On the other hand, and as mentioned in our Discussion section, patient-related caries risk and occlusal load may indeed influence restoration survival and success (Collares 2016; vande Sande 2016; Archibald 2017). This is probably – although not significantly – apparent from the flow chart, showing loss of 3 restorations in 1 patient due to caries. The variables at the patient level in se are not directly related to the restoration’s physicomechanical properties.
- The flow of participants through the phases of the study is missing and needs to be presented urgently and with full details.
We added a flow chart (figure 1) on patient and restoration numbers.
- There is now agreement in the literature that two evaluators independently assess the chosen criteria. In case of different evaluations, both then agree on a common value - still with the patient on the chair. Why was this not done?
For practical reasons, assessment by two evaluators was not possible in this educational clinical setting. Only one evaluator was available for the follow up of this clinical study. However, calibration of the evaluator was done prior to the start of the study.
- The authors mention at various points in the manuscript that the statements made are sufficiently reliable, but refrain from a power analysis. Please do an adequate power analysis and report these results.
I am afraid a post-hoc power analysis on this clinical study will not give us any extra information as described by Zhang Y. 2019, however, as described in an earlier answer, the number of restorations included in the study was depended on the amount of available undergraduate students of that year. However if you still want us to add this analysis, we will still be available for you.
- The first follow-up examinations took place after 14 months, the latest after 44 months. Considering that in similar studies follow-up intervals of 6 or 12 months are usual, this range is much too long. Appropriate periods should be defined here, e.g. first year, 2nd year, etc. and the data analysed accordingly.
We understand that the usually applied follow-up periods for scoring restorations are 6, 12, 24 months etc.…The same practical reasons as specified in our answer to remark #6, together with logistical restrictions and limited availability of the patients and examiner due to the Covid-19 pandemic, are responsible for this unusual time frame. Although this might obscure comparison with literature, we do not believe that this has a dominant effect on the reported clinical outcomes. A quick analysis of the sub-scores as presented in the bar graph of Figure 2 shows that the small numbers do not allow statistically valid conclusions to be drawn.
- “Up to 44 months ….” in the title is completely misleading.
We now realize that, despite our efforts to inform the reader about the maximal stretch of the evaluation period, the phrasing indeed might be confusing. We hence choose to omit “Up to 44 months” from the title. The title is now presented as a completely new one, holding in account all reviewers’ comments and suggestions.
- Statistical evaluation
- A normality test on ordinal scaled USPHS data is false.
- Present all USPHS data as frequencies including appropriate percentages.
- Apply chi-square tests on USPHS data.
- The VAS results clearly show that these are not symmetrically distributed. Therefore, they cannot be normally distributed either. So it is necessary to present and evaluate these data non-parametrically. The representation as in Fig 2 is incomplete.
- Yes, indeed, we now recognize that this test was wrongly used, and it was therefore removed from the text.
- The USPHS data are already presented as N(%) in Table 5.
- The chi-square test results on USPHS-data are added to table 5 in an extra column. They show the same results as the Kruskall-wallis test but do not show the in-depth differences between different types of restorations.
- The representation in figure 2 is that of the median and interquartile ranges and is therefore a non-parametrical representation as you recommended. A clarification of this has been added to the text and figure description. As medians are rarely used in dentistry, we preferred to describe the means in the text.
- Include manufacturers informations, Lot numbers, and all further details throughout the manauscript.
We cross-checked the manuscript and added additional information where needed. All restorations were outsourced for fabrication and the LOT numbers unfortunately were not provided by the lab. We agree that adding these numbers might have added to the traceability of the process but, on the other hand, we feel that not mentioning them does not diminish the value of the message of the manuscript.
- The parts of the manuscript do not coincide, e.g., information to be presented in M&M is presented only in the Discussion (e.g.: exclusion of patients with high caries risk). Structure the manuscript very clearly.
Thank you for your valuable comment; we adapted the M&M section accordingly.
- The unit of investigation was not chosen correctly. Therefore, it is not possible to review the results and their discussion.
As mentioned in the reply to question 4, choosing the patient as the investigation would presumably not change the restoration outcomes. On the contrary, many restoration-related variables are specific for the restoration’s behavior and are independent of the patient, except for a possible effect of an individually elevated caries risk on restoration survival. Therefore, choosing only one restoration per patient would obscure the results.
- The discussion of the material used and the methods applied is not exhausting.
We agree that a more in-depth or comprehensive discussion of materials and methods would enrich the manuscript. We tried however to stay in line of previous articles on the topic and referred to basic reference articles where to find more information on materials and methods.

Reviewer 2 Report
thanks for the article and I have the following comments:
1. change the title"Up to 44 months clinical evaluation of nano-ceramic Computer Assisted Design/Computer Assisted Machining restorations 3 placed by undergraduate students" to "Placement of CAD/CAM resin composite restorations by undergraduate students: 44-month clinical evaluation"
2. "nano ceramic" is the Lava Ultimate commercial gimmick. Please change all "nano ceramic" into other names, such as "resin composite" Please also change some of the text contents so that you can differentiate the indirect restoration resin composite blocks and direct restoration of curable resin composite.
3. CAD-CAM --> change all to CAD/CAM
4. p2L75 "It was reported that these resin-based materials, referred to as hybrid ceramics or nano-ceramics..." First, the "resin-based" should change it to "resin-containing", because the hybrid ceramics like Enamic has two continuous phases (ceramics and polymer), but resin composites like Lava Ultimate and Cerasmart are simply resin composites that cannot be named as ceramic! So, I may suggest you change this sentence to "It was reported that these resin-containing CAD/CAM materials, referred to as hybrid ceramics (e.g. Enamic) or resin composite (e.g. LAVA Ultimate and Cerasmart)..."
5. The article is worthy but the emphasis should be put the easiness and efficient of CAD/CAM indirect restoration placement by undergrad who has comparatively less clinical experience. So, in the Introduction and discussion, please rewrite and making the emphasis about learning curve, CAD/CAM and digital dentistry in dental education, and add some reference.
6. What about silane? I bet the Lava Ultimate bond better using silane primer but you have not use it..?
Author Response
Response to reviewer 2
To Reviewer # 2: First of all, we want to thank you for your insightful and useful comments. We tried to address all your comments in the best way we could and we hope that the reviewed version of the manuscript has improved. We are still fully available to modify the text if again prompted.
- change the title"Up to 44 months clinical evaluation of nano-ceramic Computer Assisted Design/Computer Assisted Machining restorations 3 placed by undergraduate students" to "Placement of CAD/CAM resin composite restorations by undergraduate students: 44-month clinical evaluation"
We have changed the title according to your and reviewer 1’s recommendations to the following: “Clinical evaluation of resin composite CAD/CAM restorations placed by undergraduate students”
- "nano ceramic" is the Lava Ultimate commercial gimmick. Please change all "nano ceramic" into other names, such as "resin composite" Please also change some of the text contents so that you can differentiate the indirect restoration resin composite blocks and direct restoration of curable resin composite.
Thank you for your valuable comment, this has been adjusted accordingly throughout the manuscript. We changed “nano-ceramic” to “resin composite (CAD/CAM)“
- CAD-CAM --> change all to CAD/CAM
We initially used CAD-CAM as defined in the ninth edition of “the glossary of prosthodontic terms” by the Academy of Prosthodontics. However, as per your suggestion, all “CAD-CAM” acronyms have been changed to “CAD/CAM”.
- p2L75 "It was reported that these resin-based materials, referred to as hybrid ceramics or nano-ceramics..." First, the "resin-based" should change it to "resin-containing", because the hybrid ceramics like Enamic has two continuous phases (ceramics and polymer), but resin composites like Lava Ultimate and Cerasmart are simply resin composites that cannot be named as ceramic! So, I may suggest you change this sentence to "It was reported that these resin-containing CAD/CAM materials, referred to as hybrid ceramics (e.g. Enamic) or resin composite (e.g. LAVA Ultimate and Cerasmart)..."
Thank you for your valuable comment, this has been inserted in the manuscript as suggested.
- The article is worthy but the emphasis should be put the easiness and efficient of CAD/CAM indirect restoration placement by undergrad who has comparatively less clinical experience. So, in the Introduction and discussion, please rewrite and making the emphasis about learning curve, CAD/CAM
This is indeed a point of view which we tried to put more in the spotlight, therefor we had included some paragraphs about learning curve in undergraduate students, but more text and references were added in introduction and discussion (lines 52-70;408-432) as per your suggestion, hoping you will find this sufficient.
- What about silane? I bet the Lava Ultimate bond better using silane primer but you have not use it..?
There is a lot of controversy about this in the literature. Peumans M. et al. reported in 2016 that HF application in combination with Silane produces the highest bond strength with a third-party adhesive. However another study showed a decrease of bond strength when silane was used, and had the highest bond strength when the manufacturers workflow was followed. (Emserman I. 2019) In this study we chose to follow the workflow as proposed by the manufacturer (3M) which uses sandblasting in combination with the silane and 10-MDP containing Scotchbond Universal as pretreatment.

Reviewer 3 Report
Manuscript ID: jcm-1273643
“Up to 44 months clinical evaluation of nano-ceramic Computer Assisted Design/Computer Assisted Machining restorations placed by undergraduate students.”
The aim of this clinical study was to follow up CAD/CAM milled restorations made of a highly filled composite resin (Lava Ultimate) after 3.5 years.
The study is very well conducted and the manuscript is very well written. Therefore, only minimal corrections are suggested. The authors refer to the Lava Ultimate material as "nano-ceramic" throughout the manuscript. I find this confusing and incorrect. To the best of my knowledge, Lava Ultimate is a composite resin filled with approximately 80% by mass of nano-ceramic particles. It is therefore not a ceramic block from which inlays or onlays have been milled. The term should therefore be changed throughout the manuscript.
There are also minor spelling mistakes. For example, it is sometimes United States, then USA again. I would write something like that consistently in the manuscript.
Author Response
Response to reviewer 3
To Reviewer #3: First of all, we want to thank you for your insightful and useful comments. We tried to address all your comments in the best way we could, and we hope that the reviewed version of the manuscript has improved. We are still fully available to modify the text if again prompted.
The aim of this clinical study was to follow up CAD/CAM milled restorations made of a highly filled composite resin (Lava Ultimate) after 3.5 years.
The study is very well conducted and the manuscript is very well written. Therefore, only minimal corrections are suggested. The authors refer to the Lava Ultimate material as "nano-ceramic" throughout the manuscript. I find this confusing and incorrect. To the best of my knowledge, Lava Ultimate is a composite resin filled with approximately 80% by mass of nano-ceramic particles. It is therefore not a ceramic block from which inlays or onlays have been milled. The term should therefore be changed throughout the manuscript.
Thank you for your valuable comment, this has been adjusted accordingly throughout the manuscript. We changed “nano-ceramic” to “resin composite (CAD/CAM)“
There are also minor spelling mistakes. For example, it is sometimes United States, then USA again. I would write something like that consistently in the manuscript.
The term USA was replaced by United States and a thorough spell check has been reconducted.
